# Enhanced Manifold of States Achieved in Heterostructures of Iron Selenide and Boron-Doped Graphene

Valentina Cantatore and Itai Panas *

Department of Chemistry and Chemical Engineering, Energy & Materials, Chalmers University of Technology, Gothenburg 41296, Sweden; valcan@chalmers.se
* Correspondence: itai.panas@chalmers.se; Tel.: +46-31-7722860

**Abstract:** Enhanced superconductivity is sought by employing heterostructures composed of boron-doped graphene and iron selenide. Build-up of a composite manifold of near-degenerate noninteracting states formed by coupling top-of-valence-band states of FeSe to bottom-of-conduction-band states of boron-doped graphene is demonstrated. Intra- and intersubsystem excitons are explored by means of density functional theory in order to articulate a normal state from which superconductivity may emerge. The results are discussed in the context of electron correlation in general and multi-band superconductivity in particular.

**Keywords:** superconductivity; boron-doping; graphene; FeSe; electron correlation; heterostructures

## 1. Introduction

The requirements of quasi two-dimensional (2D) materials to be stable implies satisfying two main conditions, each offering opportunities in physical and chemical engineering. Firstly, the self-sufficient termination of the electronic structure enforced by its implied robustness—including e.g., semi-metallic properties—may be explored by means of doping. Secondly, a spatial third dimension is accessible for catalysis, sensor, and/or protection applications. The present work attempts to contribute to a third line, comprising the quest for individually symmetry protected narrow-gapped 2D subsystems, stacked along the third spatial direction employing "spacers", so-called buffer layers, to form hetero-structures. Here, tailored electronic heterostructures are sought, ambitiously, beneficial to superconductivity. Indeed, this approach reminds of emerging superconductivity at the interface between $LaAlO_3$ and $SrTO_3$ (LAO/STO), each of which being an insulator [1]. The quest for improved superconducting properties by combining superconducting and nonsuperconducting components is not new. Indeed, e.g., $Au_2Bi$ is a superconductor at ambient conditions while neither Au(s) nor Bi(s) is a superconductor [2]. Even more relevant here are the multilayer cuprates. While only the outer layers in the stacks in the unit cell are superconducting, yet, optimal Tc (the critical temperature for superconductivity) is achieved with three layers per unit cell in e.g., the Hg cuprates [3]. Enhanced density of states at the Fermi level ($E_F$ from now on) as manifested in inter-plane charge transfer has been suggested to explain the observation, i.e., owing to partial densities of states (PDOS) of conducting hole band in the outer planes overlapping with the PDOS of the valence band of the inner plane [4].

We work on an understanding of superconductivity which is generic with respect to local and nonlocal nonadiabaticity where the latter produces superconductivity. Thus, at the heart of superconductivity is the degeneracy among noninteracting reference states. Equally central is the Jahn–Teller theorem telling of the spontaneous lifting of electronic degeneracy by structural symmetry breaking. In the solid state this is manifested in the Peierls instability. However, often use can be

made of the a priori high density of states at the Fermi level to heal the symmetry broken state. This requires coupling of irreducible representations of the local point group symmetry, i.e., symmetry violating virtual excitations resulting in Bogoljubov quasiparticles [5]. Thus, a correlated many-body ground state is accessed by means of virtual excitations among the available eigenstates of the relevant noninteracting Hamiltonian. A von Neumann entropy contribution to the electronic free energy is then obtained by mixing nominally near-degenerate noninteracting states by means of suitable field particles mediating the coupling. These may constitute virtual photons, phonons, magnons, etc., protecting against symmetry breaking. The cooperative access of virtual excitations however, is conditioned by global electronic phase coherence, the hallmark of superconductivity.

In pure metals the noninteracting electronic Hamiltonian of the superconductor is represented by its density of states at the Fermi level, $DOS(E_F)$, where non-adiabatic electron–phonon coupling is required in order to stay at the structural instability point, thus transforming the material into a superconductor along the line of Bardeen-Cooper-Schrieffer theory [6]. In contrast, the quasi-degenerate manifold of electronic embedded atom states belonging to different irreducible representations constitutes the essential component of a multiband superconductor, the virtual excitations of each acting mutual field particle and charge carrier. Aspects of a particular Hund's rule violating magnon–exciton mediated entanglement of a priori noninteracting manifolds of states was claimed relevant also to the superconducting cuprates [7,8]. For the cuprate superconductors we have proposed a symmetry protected two-gapped five-bands-model of local $\sigma$ and $\pi$ symmetries relative to the Cu–O–Cu axis to constitute a normal state from which superconductivity emerges [4,7–9]. The zeroth order Hamiltonian was taken to reflect two narrow gap semiconductors displaying antiferromagnetic and checkerboard charge density wave instabilities and becoming entangled by virtual magnons–excitons in each subsystem of each individual 2D $CuO_2$ plane. Two dimensional superconductivity emerges from the equally unifying grand canonical ensembles of the two subsystems owing to the common chemical potential—i.e., the common $E_F$—of the two subsystems [4] offering intersubsystem transfer of Cooper pairs [10] analogous to Josephson coupling [11]. The effective three dimensional (3D) phenomenon in turn results from such Josephson coupling owing to the common $E_F$ in the stacks $CuO_2$ planes which constitute an essential part of the solid.

The present work is motivated by the search for higher $T_C$ in the FeSe system [12] and is relevant for the emergence of topological superconductivity in graphene and related materials utilizing e.g., Majorana bound states [13,14]. We envisage utilization of the pseudo-gap observed for functionalized boron-doped graphene (BBG from now on) [15] as well as for BBG@$Cu_{111}$ [16]. Indeed, this pseudogapped ground state is understood to exhibit a correlated ensemble of virtual particle–hole excitations out of the vacuum reference state. Such additional Majorana resonances in turn contribute to the resulting effective ground state, thus adding to its robustness. Moreover, in analogy to the interplane excitations' contribution to the superconducting ground state in the triple layered $Hg_{1223}$ [4], we seek to enhance the $DOS(E_F)$ of the resulting superconductor, here composed of an FeSe and boron-doped graphene heterostructure (see Figure 1), in order to enhance the Hilbert space of a priori degenerate eigenstates of $H_0$ subject to entanglement. While electron–phonon and exciton–magnon couplings do indeed offer ways to access the enhanced superposition of virtual states that contribute to the correlated effective ground state that is the superconducting state, the present study avoids issues related to coupling altogether while focusing on the actual enhancement of the manifold of near-degenerate states.

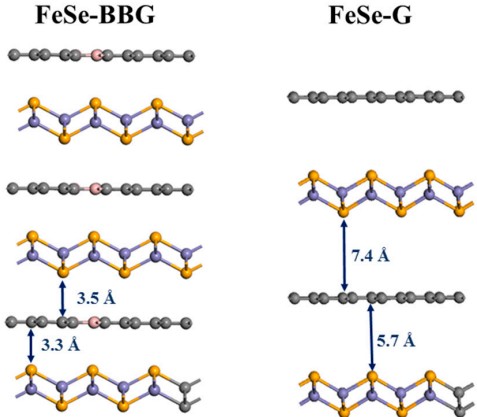

**Figure 1.** Samples of FeSe-BBG (**A**) and FeSe-G (**B**) heterostructures. Color code: grey, carbon; pink, boron; yellow, selenium; light blue, iron. Arrows indicate the inter- and intra-cell distances (see text).

## 2. Theoretical Background

The entanglement of the a priori degenerate eigenstates of $H_0$ may be formulated by using an analogy to Fock space to treat explicit electron correlation by the variational configuration interaction (CI) formalism. Here, the ground state energy for the wave function takes the form

$$
\begin{aligned}
E &= \sum_{M,N}^{\substack{\text{degenerate}\\\text{states}}} \left[ c_M^* c_N \langle \Psi_M | \hat{H} | \Psi_N \rangle \right] \\
&= \sum_{M,N}^{\substack{\text{degenerate}\\\text{states}}} \left[ c_M^* c_N \left\{ \sum_{i,j} h_{ij} \langle M | \hat{E}_{ij} | N \rangle + \tfrac{1}{2} \sum_{i,j,k,l} g_{ijkl} \langle M | \hat{E}_{ij}\hat{E}_{kl} - \delta_{jk}\hat{E}_{il} | N \rangle \right\} \right]
\end{aligned}
\tag{1}
$$

where $\hat{E}_{ij}(= \hat{a}_{i\alpha}^\dagger \hat{a}_{j\alpha} + \hat{a}_{i\beta}^\dagger \hat{a}_{j\beta})$ are spin summed creation and annihilation operators for virtual exciton [1]. The second (two-electron) term in Equation (1) couples so-called configuration state functions (CSF), $|M\rangle$ and $|N\rangle$, comprising spin adapted linear combinations of Slater determinants that differ by two virtual excitons $j \rightarrow i$ and $l \rightarrow k$. Taking a single Slater determinant to reflect the vacuum state the first (one-electron) term in brackets is removed if virtual single-excitons are not allowed to couple to the virtual double-exciton states, i.e., configuration interaction including only doubles (CID). Moreover, single-excitons do not couple to the chosen vacuum state due to Brillouin's theorem. Such omission is equivalent to disallowing quasi-particle excitations to directly contribute to the correlated ground state. This way, the double-sum over electron configurations is transformed into a single-sum over double-exciton CSF:s. Now, for two electronic subsystems, we partition Equation (1)

$$
\begin{aligned}
E &= \sum_{N=0}^{\substack{\text{degenerate}\\\text{exciton pairs}}} \left[ c_0^* c_N \langle 0 | \hat{H} | N \rangle \right] \\
&= \sum_{N=0}^{\substack{\text{degenerate}\\\text{exciton pairs}}} c_0^* c_N \left\{ \sum_{\{i,j,k,l\}\epsilon A} \tfrac{1}{2} g_{kl\epsilon A}^{ij\epsilon A} \langle 0 | \hat{E}_{ij}\hat{E}_{kl} - \delta_{jk}\hat{E}_{il} | N \rangle \right. \\
&\quad \left. + \sum_{\{i,j,k,l\}\epsilon B} \tfrac{1}{2} g_{kl}^{ij} \langle 0 | \hat{E}_{ij}\hat{E}_{kl} - \delta_{jk}\hat{E}_{il} | N \rangle + \sum_{\substack{\{i,j\}\epsilon A\\\{k,l\}\epsilon B}} g_{kl}^{ij} [\langle 0 | \hat{E}_{ij}\hat{E}_{kl} | N \rangle + \langle 0 | \hat{E}_{il}\hat{E}_{kj} | N \rangle] \right\}
\end{aligned}
\tag{2}
$$

to obtain apparent virtual single-exciton terms emerging in each subsystem as a consequence of the coupling to the other subsystem, i.e., third term in Equation (2). Thus, our analogy is complete when noting that virtual excitons $\{[j \rightarrow i]\epsilon A \, ; \, [l \rightarrow k]\epsilon B\}$ and $\{[j_A \rightarrow k_B] \, ; \, [l_B \rightarrow i_A]\}$, where $A$ and $B$ refer to the two subsystems, reflect superpositions of apparently incompatible virtual single-exciton states, which become accessible by the mutual entanglement. This particular all-electronic nonadiabaticity—i.e., between excitons coupling two different subsystems each acting embedding for the other—has been formulated by us previously. Indeed, there is a general consensus in the community regarding the multiband origin of superconductivity in FeSe(s) already within the FeSe layers owing to entanglement among the Fe 3d-bands present at the Fermi energy [8]. The generic nature of the above mapping of CID on the superconducting ground state is further emphasized as we understand 3D superconductivity in FeSe(s) to be again achieved by intersubsystem virtual exciton–exciton coupling, and this time between single FeSe layers. Intercalations by neutral molecules as well as by salts has offered a means to raise the critical temperature beyond that of pure FeSe [17]. Complementary to the introduction of spacers and external ionic interactions, the present study emphasizes the potential of electronically active intercalants as manifested by the boron-doped graphene.

It is furthermore emphasized that coexistence of two or more mutually nonhybridizing bands at the same chemical potential is at the heart of our multiband understanding of superconductivity. In as much as the Fermi energy is identical with the electronic chemical potential, the matching $E_F$:s of different components, as estimated from Kohn–Sham density functional theory (KS-DFT), is attempted in search for candidate heterostructures offering candidate superconductors. Acknowledging the shortcomings of KS-DFT, in the present study we test the possible relevance of the emerging Fermi energy matching. Thus, rather than absolute numbers, such matchings of $E_F$:s are taken to rely on intersubsystem excitation energies taking small numerical values.

## 3. Computational Considerations

Our calculations have been performed using the CASTEP 7.0 code [18] with ultrasoft pseudopotentials. A kinetic energy cut-off of 300 eV was used for the plane waves in the basis set together with a $2 \times 2 \times 2$ Monkhorst-Pack $k$-point mesh. The adsorption of the graphene (or boron-doped graphene) on the FeSe surface has been modeled using the dispersion-corrected DFT-D2 scheme [19,20] with the PBE functional (PBE-D2). Conceptual understanding is extracted from the computed electronic density of states, DOS, as obtained by KS-DFT. In this context, it is noted that said London interactions contribute an energy term, which is inaccessible to KS-DFT as the latter constitutes a single Slater determinantal wave function. This implies that said London interactions do not directly affect the Kohn–Sham density of states. They are crucial, however, to the cohesion between subsystems and thereby they indirectly impact the DOS via the resulting structural modifications. This is what is taken into account in the present study. The importance of proximity is emphasized by the intersubsystem charge transfer made accessible in the triplet excited states.

Moreover, we have used the atomic-limit DFT + U approach, in order to be able to treat strong on-site electron–electron correlations among the Fe-d electrons explicitly through an additional Hubbard-U term, where U is the on-site Coulomb interaction parameter [21,22] that we set at the value of 2.5 eV.

Intercalation of single-layer boron doped graphene in between every layer of FeSe is considered exclusively in order to focus on the electronic impacts resulting from the hetero-structure. Any emerging enhancement of near-degenerate manifold of states owing to interfacing the two subsystems presented below, would be diluted by the interventions of any subsystem multilayering, irrespective of boron-doped graphene or iron selenide. Along a similar line, while copper has been demonstrated to constitute a viable candidate for supporting boron-doped graphene thus offering both electronic and thermal contact [16,23], the bulk properties of the infinite stack of single-layer subsystems [-FeSe–BG-]inf, is understood to be independent of substrate.

Employing a supercell subject to periodic boundary conditions in order to learn of properties of heterostructures implies dealing with incommensurate subsystems. Enforcing the superstructure renders the softer system exposed to a stress-strain field owing to boundary conditions set by the stiffer subsystem. The impact of stress on multiorbital superconductivity in general and in the iron based superconductors in particular has been discussed previously [24]. Rather than being detrimental to the superconductivity it was shown how the phenomenon could even be even promoted by stress. Here, in order to obtain the best epitaxy between the two layers we used a $2 \times 3$ supercell of FeSe on top of which we introduce a graphene (or boron-doped) layer in the form of a $3 \times 5$ supercell. The systems considered were thus subjected to the periodic boundary conditions and atomic positions and cells parameters were optimized with the BFGS algorithm using delocalized internal coordinates. However, only vertical singlet-triplet excitations were considered. Thus, the cell parameters become a = 7.42 Å, b = 12.58 Å and c = 14.43 Å in case of graphene-FeSe, and a = 7.41 Å, b = 12.63 Å and c = 9.08 Å in case of boron-doped graphene-FeSe. The distances obtained between FeSe and the graphene plane is of 7.4 Å intra-cell and 5.7 Å inter-cell while the distance between FeSe and the BBG layer is of 3.3 Å intra-cell and 3.5 Å inter-cell (Figure 1).

## 4. Results and Discussion

Arguably, the cuprate superconductors as well as the iron pnictides offer two heterostructured classes. A particularly straightforward candidate of the latter is ThFeAsN [25] which is composed of two ionic subsystems, i.e., $(ThN)^+$ and $(FeAs)^-$. This is in contrast to the FeSe superconductor which is a layered quasi-2D solid, composed of electronically neutral planes analogous to graphite. Successful attempts at achieving tunability in the FeSe system include—besides pressure—neutral molecular intercalants as well as alkali metal organic [26] and molecular [27]. Attempts at merging of graphene with FeSe were made previously. Thus, the floating phenomenon in case of FeSe on graphene [28] is taken as starting point in the present investigation.

Here, we explore the potential of an FeSe + boron-doped graphene composite for tuning the electronic properties of the resulting hetero-structure (see Figure 1 again). Use is made of the opening of a pseudogap in boron-doped graphene. This occurs in freestanding BBG when subject to molecular Lewis acid-base type dative bonding to the boron sites residing in different graphene sublattices. Our effort extends on previous studies demonstrating universal socket-plug coupling employing dative bonding between incoming lone-pair on nitrogen and the B $2p_z$ orbital for sensor and catalysis applications [15,29]. Moreover, cohesion between $Cu_{111}$ surface and boron doped graphene was previously demonstrated, the $Cu_{111}$ surface acting electron donor towards the B $2p_z$ [16]. On $Cu_{111}$ supported BBG, intersubsystem RKKY type coupling lifts the different sublattices constraint for the opening of the pseudogap, which was observed for freestanding BBG.

We employ density functional theory in conjunction with Hubbard U to arrive at an antiferromagnetic AFM state of FeSe, i.e., $FeSe_{AFM}$, which we take to represent the electron structure of the FeSe ground state, i.e., modeling the reference (vacuum) state, see Figure 2c.

It is noted that while densities of $\alpha$ and $\beta$ spin states are identical, the FeSe subsystem is antiferromagnetic while the boron-doped graphene is nonmagnetic. Similar to the $BBG@Cu_{111}$ case, the pseudogap in BBG when bound to FeSe results partially from the polarization of the BBG by the substrate. Indeed, it is the resulting match between the conduction band of pseudogapped BBG and the valence band of the $FeSe_{AFM}$ substrate which allows the partial electron transfer from FeSe into BBG and also the buildup of a composite density of states at the Fermi energy, see Figure 2 again. Besides the van der Waals interaction, the two contributions, acting to improve cohesion between the two components in the resulting heterostructure, are the reduced intersubsystem Pauli repulsion and corresponding partial charge transfer analogous to dative bonding in molecules. It is emphasized that proximity between the boron doped graphene and FeSe subsystems is a necessary prerequisite for intersystem excitonic couplings. In particular, two boron atoms separated by a $C_2$ unit [B\=/B] and by a $C_4$ unit [B\=/=\B] in a graphene superlattice were considered. In the case of the former, the pure

vacuum states were found to prevail, i.e., [FeSe]$_{AFM}$ and [B\=/B]$_{PG}$, PG referring to the pseudogap in BBG. In order to validate the implied charge transfer instability manifested in the ground state PDOS of [FeSe]$_{AFM}$-[B\=/B]$_{PG}$, the ferromagnetic triplet state was resorted to, see Figure 3. It is gratifying to note that the singlet–triplet excitation comes out at +0.08 eV supporting the implied near degeneracy emerging from the DOS of the singlet (see Figure 2a).

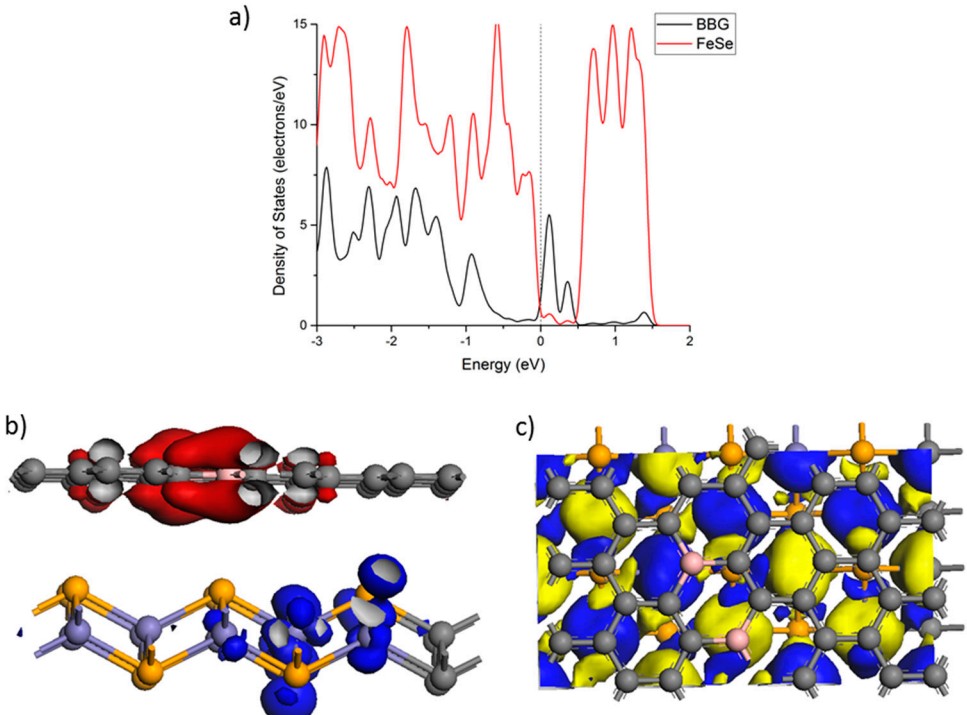

**Figure 2.** (**a**) The electronic partial densities of states PDOS of the [FeSe]$_{AFM}$-[B\=/B]$_{PG}$ display near-degeneracy at E$_F$. While each subsystem is seen to be gapped, the near-degeneracy is owing to the proximity of the top of the valence band (T-VB) of [FeSe]$_{AFM}$ to the bottom of the conduction band (B-CB) of [B\=/B]$_{PG}$. (**b**) Characteristic T-VB (blue) and B-CB (red) Kohn–Sham states, respectively. (**c**) Spin density is seen to reside solely on the [FeSe]$_{AFM}$ subsystem. Blue/yellow: +/−.

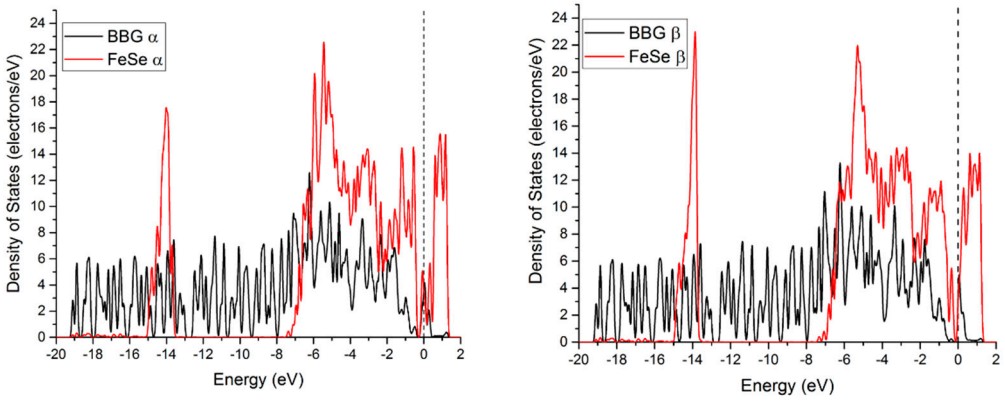

**Figure 3.** The α- and β-spin PDOS of the triplet state in the supercell telling of hole doping of the FeSe subsystem and electron doping of the [B\=/B]$_{PG}$ subsystem.

It is noted how the FeSe associated gap is filled owing to the hole doping of the FeSe subsystem. Crucially, the hole state *h* takes selenium character to a significant extent, while the particle state *e* resides on the [B\=/B], see also Figure 4. The drastic rebuilding of the electronic structure, resulting in

the [FeSe]$^h_{AFM}$, reflects a hole doped antiferromagnet. It is contrasted by the minor impact on PDOS of [B\=/B]$^e$ of electron transfer as its conduction band becomes occupied.

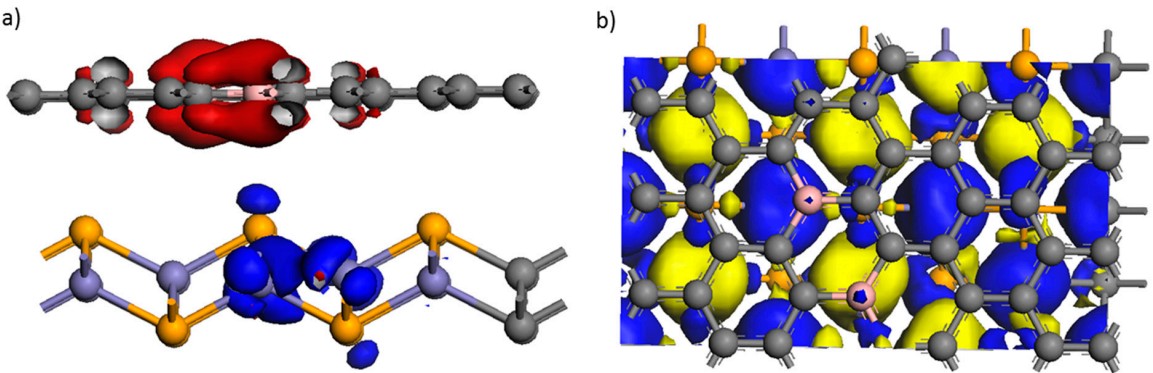

**Figure 4.** (**a**) Two states residing at E$_F$ of α-spin character, hole state (blue) of [FeSe]$^h_{AFM}$, and electron state (red) of [B\=/B]$^e$. (**b**) resulting spin density of [FeSe]$^h$-[B\=/B]$^e$. Magnetic structure is seen to change (compare Figure 3) reflecting the [FeSe]$^h_{AFM}$, while spurious spin on the boron doped graphene manifests the [B\=/B]$^e$ electron doping of this subsystem.

The dependence of electronic response to the separation between boron atoms in BBG is shown in Figure 5.

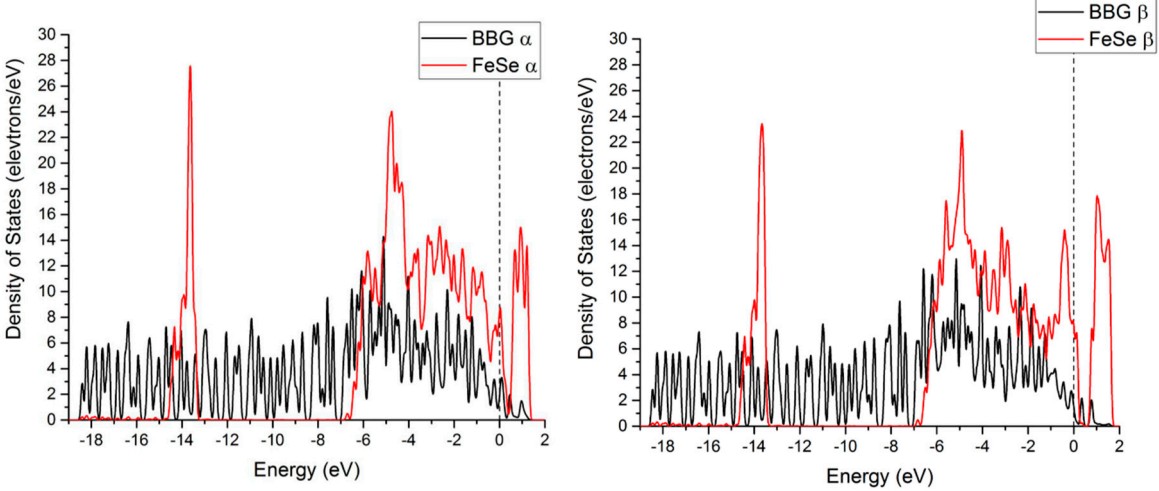

**Figure 5.** The α- and β-spin PDOS of the singlet state in the [FeSe]-[B\=/=\B] supercell, telling of hole doping of the [FeSe] subsystem and electron doping of the [B\=/=\B] subsystem already for the ground state, cf. Figure 4.

Indeed, already the singlet ground state of [FeSe]-[B\=/=\B] displays charge transfer resulting in [FeSe]$^h$-[B\=/=\B]$^e$, see also Figure 6. In contrast to the [FeSe]$^h$-[B\=/B]$^e$ in its triplet excited state, see Figure 4, the electron state in the [B\=/=\B]$^e$ subsystem is delocalized. Again, near-degeneracy implied from the DOS (Figure 5) is consistent with the singlet–triplet excitation is found to be at +0.38 eV reflecting an intraplane FeSe excitation as access to the interplane charge transfer is prohibited by the acceptor state of [B\=/B]$^e$ already being occupied in the ground state. Thus, the computed excitation refers to a doublet–quartet excitation in the [FeSe]$^h$ subsystem.

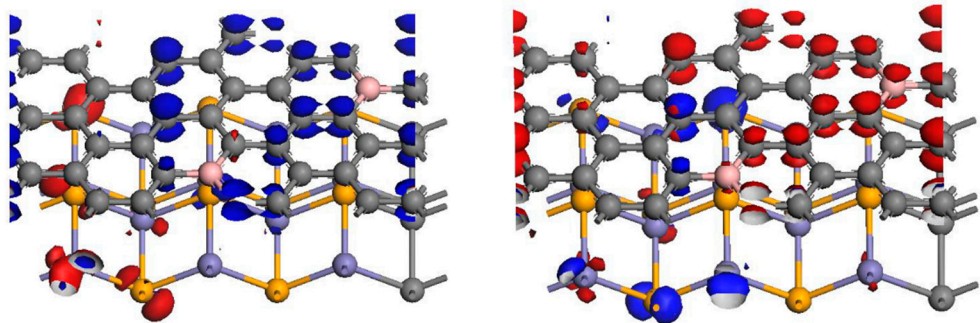

**Figure 6.** Kohn–Sham states at EF for the ground state [FeSe]h-[B\=/=\B]e. The delocalized character in the [B\=/=\B]e emphasizes its metallic character. This is in contrast to [FeSe]h-[B\=/B]e where the [B\=/B]e electron state is localized, cf. Figure 3.

These intersubsystem excitations discussed for $[FeSe]^h$-$[B\backslash=/B]^e$ and $[FeSe]$-$[B\backslash=/=\backslash B]$ are understood to be representative of virtual excitations among degenerate pure-states contributing to a resulting entangled ground state along the lines of the double-exciton term (cf. Equation (2))

$$\sum_{N=0}^{\substack{degenerate \\ exciton\ pairs}} \sum_{\substack{\{i,j\}\epsilon BBG \\ \{k,l\}\epsilon FeSe}} g_{kl}^{ij}\langle 0|\hat{E}_{il}\hat{E}_{kj}|N\rangle \tag{3}$$

Besides these excitations, there are the virtual intrasubsystem excitons projected out by the Hubbard U term in the DFT calculations. Accidentally, a representative of this set of virtual excitons i.e.,

$$\sum_{N=0}^{\substack{degenerate \\ exciton\ pairs}} c_0^* c_N \left\{ \sum_{\{i,j,k,l\}\epsilon FeSe} \frac{1}{2} g_{kl\epsilon A}^{ij\epsilon A}\langle 0|\hat{E}_{ij}\hat{E}_{kl} - \delta_{jk}\hat{E}_{il}|N\rangle \right\} \tag{4}$$

as well as one of the intersubsystem coupled intrasubsystem excitons in

$$\sum_{N=0}^{\substack{degenerate \\ exciton\ pairs}} \sum_{\substack{\{i,j\}\epsilon FeSe(A) \\ \{k,l\}\epsilon FeSe(B)}} g_{kl}^{ij}\langle 0|\hat{E}_{il}\hat{E}_{kj}|N\rangle \tag{5}$$

is found for one of the FeSe planes in a supercell calculation including two [FeSe]-[B\=/B] double-planes in the superlattice, see Figure 7.

Thus, the particle–hole pair is understood to reside on one of the FeSe planes, reflecting a $[FeSe]^{h-e}{}_{AFM}$ configuration, while the BBG planes remain undoped as displayed in Figure 8.

While terms on the form of equation (5) may contribute to the correlated ground state in FeSe(s), it is possible that coupling via intervening BBG planes will indeed enhance the corresponding resonances.

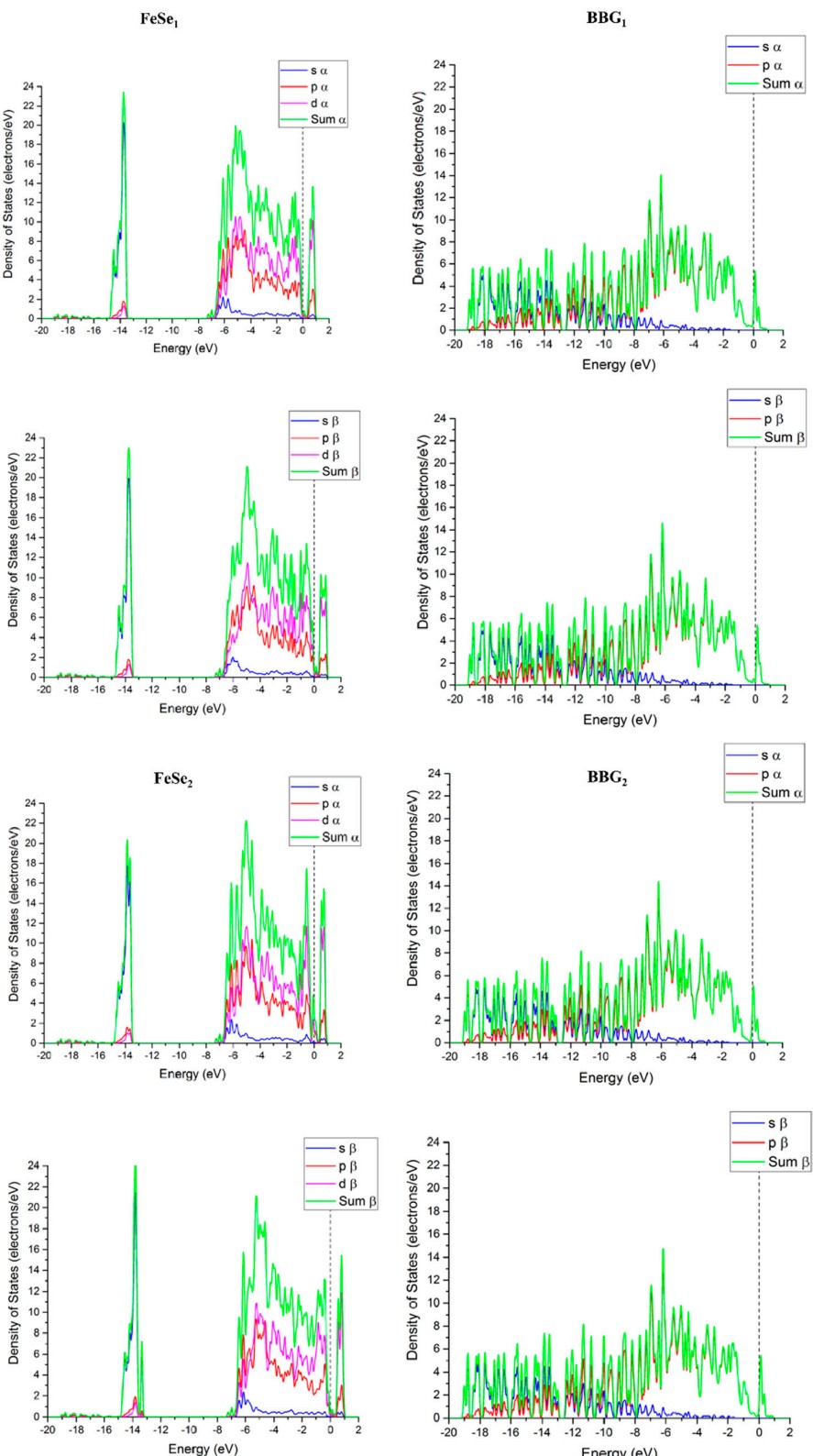

**Figure 7.** Results for a supercell calculation including two [FeSe]-[B\=/B] double-planes in the superlattice. **Right panel**: PDOS of the two [B\=/B] subsystems telling of B-CB residing just above $E_F$, consistent with Figure 2a, for both planes. **Left panel**: PDOS of the two [FeSe] subsystems. Top two graphs show same [FeSe]$_{AFM}$ PDOS as in Figure 2a, while the second two graphs tell of a magnetic excitation in the corresponding [FeSe]$_{AFM}$ plane, i.e., [FeSe]$^{h-e}_{AFM}$.

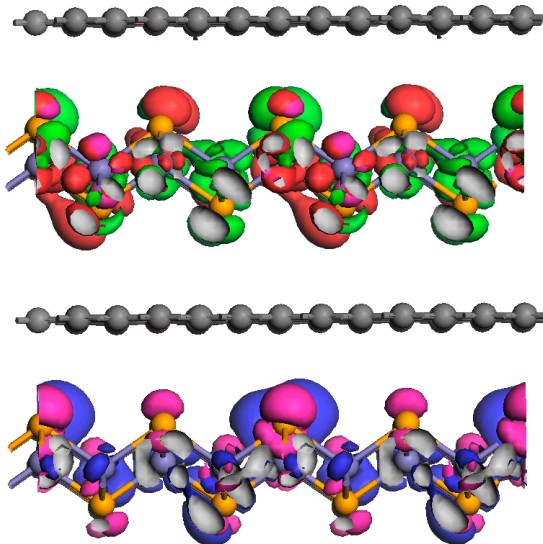

**Figure 8.** Four Kohn–Sham states below $E_F$ for a supercell calculation including two [FeSe]-[B\=/B] double-planes in the superlattice. While the T-VB state resides on the [FeSe]$^{h-e}_{AFM}$ plane, the state just below T-VB is accommodating the [FeSe]$_{AFM}$ plane. It is noted that the [B\=/B] are not accessed, cf. right panel in Figure 7. Color code: blue/red, HOMO $\alpha/\beta$; blue/pink, HOMO-1 $\alpha/\beta$.

## 5. Conclusions

Rational design of an electronic structure from elementary 2D building blocks jointly constituting a plausible material has been demonstrated. The issue of cohesion was addressed, i.e., that of FeSe to BBG owing to reduced intersubsystem Pauli repulsion is understood to contrast the net weak interaction between FeSe and a double layer of graphene where one has been reported to float on the other. This differential effect is indeed reproduced in the present study, see Figure 1. The design of near-degenerate electronic states at the Fermi energy was realized by matching the valence band of one of the 2D components, i.e., FeSe, with the conduction band of its partner—the pseudogaped boron-doped graphene BBG—in the resulting heterostructure. Inasmuch as we understand FeSe(s) to be essentially a 2D superconductor, the intercalation of BBG offers a way not only of tuning but also enhancing the phenomenon by the enhanced density of states at the $E_F$ enlarging the Hilbert space and corresponding von Neumann entropy, i.e., the superposition of states that jointly constitute the entangled ground state. It is noted that spill-over of electrons from FeSe into BBG allows spontaneous and doping dependent buffering of charge carriers comprising holes in the FeSe subsystem, which in turn offer a means of self-tuning of the superconducting ground state to provide it with increased robustness. Beyond optimization of $T_C$ in the FeSe subsystem, indeed the resulting manifold of composite states at $E_F$ owing to the entanglement, may support effective superconductivity in the resulting heterosystem compound by its own right cf. LAO/STO. Thus, a candidate strategy for achieving superconductivity in composite materials where graphene is an active component was substantiated, albeit boron-doped.

**Acknowledgments:** We gratefully acknowledge the Swedish Research Council and Stiftelsen Olle Engkvist Byggmästare for funding as well as the Swedish National Infrastructure for Computing (SNIC) at C3SE for the computing time.

**Author Contributions:** Valentina Cantatore performed the calculations; Itai Panas wrote the paper. Both the authors are equally responsible for the paper.

**Conflicts of Interest:** The authors declare no conflict of interest.

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
