# Peer review of "Enhanced Manifold of States Achieved in Heterostructures of Iron Selenide and Boron-Doped Graphene"

_condensedmatter, doi:10.3390/condmat2040034_

Reviewer 1 Report

It is an interesting paper that encourages a material theoretically guided research of high-Tc superconductors produced by stacking together Van der Waals systems. The authors should however discuss more which could be the role of the lattice mismatch between FeSe and boron doped graphene. It is not clear if the thckness for the FeSe is of relevance for the properties of this stack predicted by the authors, could the authors be more explicit on that? Could also the authors explain more the role of the substrate and be more specific on which substrates should be used for the stack?   There is a missing figure captions on figure 7. The authors should pay attention to this and improve the quality of the picture that in the current state can't be read. Once the authors have made these improvements, I could suggest it for publication. Minor remarks: There is a typo in the abstract : "doper"

Author Response

R1: Added on line 51:

Employing a supercell subject to periodic boundary conditions in order to learn of properties of heterostructures implies dealing with incommensurate subsystems. Enforcing the superstructure renders the softer system exposed to a stress-strain field owing to boundary conditions set by the stiffer subsystem. The impact of stress on multi-orbital superconductivity in general and in the iron based superconductors in particular has been discussed previously [Kang J1, Kemper AF2, Fernandes RM1. PRL 113, 217001 (2014)]. Rather than being detrimental to the superconductivity it was shown how the phenomenon could even be even promoted by stress

R2: Added on line 43:

Intercalation of single-layer boron doped graphene in between every layer of FeSe are considered exclusively in order to focus on the electronic impacts resulting from the hetero-structure. Any emerging enhancement of near-degenerate manifold of states owing to interfacing the two subsystems presented below, would be diluted by the interventions of any subsystem multi-layering, irrespective of boron doped graphene or iron selenide. 

R3: Added on line 47:

Along a similar line, while copper has been demonstrated to constitute a viable candidate for supporting boron doped graphene thus offering both electronic and thermal contact [Cantatore, Di Valentin], the bulk properties of the infinite stack of single-layer subsystems [-FeSe--BG-]inf, is understood to be independent of substrate.

R4: Figure captions and "doper" corrected

Reviewer 2 Report

The manuscript "Enhanced manifold of states achieved in heterostructures of iron selenide and boron doped graphene", by V. Cantatore and I. Panas, seems to contain results that are interesting enough to warrant publication. However, the reading of the manuscript is made awkward by the absence of any explanation of the various symbols appearing in all the Eq.s  (1-5). The authors should also notice that there are two Eq.s numbered as Eq. (3), while Eq. (4) is missing. Finally, I dislike the identification of the chemical potential with the Fermi energy, which is not granted for interacting electrons, and requires some further assumptions (for instance, within DFT, referring to Kohn-Sham electrons).

Once the authors have taken into account the above suggestions and remarks, I think that their manuscript may be accepted for publication in Condensed Matter.

Author Response

R1: Definitions clarified in text

R2: Equation numbers corrected

R3: Added on line 117:

It is furthermore emphasised that coexistence of two or more mutually non-hybridizing bands at the same chemical potential is at the heart of our multiband understanding of superconductivity. In as much as the Fermi energy is identical with the electronic chemical potential, the matching EF:s of different components, as estimated from Kohn-Sham DFT, is attempted in search for candidate hetero-structures offering candidate superconductors. Acknowledging the shortcomings of KS-DFT, in the present study we test the possible relevance of the emerging Fermi energy matching. Thus, rather than absolute numbers, such matchings of EF:s are taken to rely on inter-subsystem excitation energies taking small numerical values.

Reviewer 3 Report

In this work the authors study an enhanced superconductivity in FeSe/B-doped graphene heterostructures using DFT calculations.
In general the study is interesting as the field of superconductivity in 2D systems is attracting a lot of interest with the hope of the realisation of high-Tc superconductors in 2D.  I would recommend the manuscript for publication, however I have some questions/comments that authors must address in the revised version:

1- The authors show an enhanced superconductivity according to the enhanced DOS in  FeSe/B-doped graphene system. However, it would be more useful if the authors evaluate the electron-phonon coupling constant and then give an estimate for Tc in this system.

2- Having a layered system, the authors correctly consider the vdW corrections into their calculations. They use dispersion‐corrected DFT‐D2 within PBE functional. Are the results for the DOS depend on the type of a vdW functional? (for example, the advanced  forms of vdW functionals (optPBE/optB86b/optB88) may change the results).  The authors should confirm (by checking for at least one case) that their results are not dependent on the choice of the vdW functional.

3- The quality of the figures (particularly the DOS figures) is poor.  The authors must enhance the quality of the figures in the revised manuscript. 

Author Response

R1: Added on line 82: 

And while e.g. electron-phonon and exciton-magonon couplings do indeed offer ways to access the enhanced superposition of virtual states that contribute to the correlated effective ground state that is the superconducting state, the present study avoids issues related to coupling altogether while focusing on the actual enhancement of the manifold of near-degenerate states.

R2: Added on line 130:

Conceptual understanding is extracted from the computed electronic density of states, DOS, as obtained by Kohn-Sham Density Functional Theory KS-DFT. In this context, it is noted that said London interactions contribute an energy term, which is inaccessible to KS-DFT as the latter constitutes a single Slater determinantal wave function. This implies that said London interactions do not directly affect the Kohn-Sham density of states. They are crucial, however, to the cohesion between subsystems and thereby they indirectly impact the DOS via the resulting structural modifications. This is what is taken into account in the present study. The importance of proximity is emphasized by the inter-subsystem charge transfer made accessible in the triplet excited states.

R3: Figure qualities have been improved.

Round  2

Reviewer 3 Report

I agree with the revised version  and I recommend the revised manuscript for publication.